# Emergence and Genomic Features of a *mcr-1 Escherichia coli* from Duck in Hungary

**DOI:** 10.3390/antibiotics12101519

**Published:** 2023-10-07

**Authors:** Ama Szmolka, Ákos Gellért, Dóra Szemerits, Fanni Rapcsák, Sándor Spisák, András Adorján

**Affiliations:** 1HUN-REN Veterinary Medical Research Institute, 1143 Budapest, Hungary; gellert.akos@vmri.hun-ren.hu (Á.G.); rapcsak.fanni93@gmail.com (F.R.); 2Department of Microbiology and Infectious Diseases, University of Veterinary Medicine, 1143 Budapest, Hungary; szemerits.dori@gmail.com (D.S.); adorjan.andras@univet.hu (A.A.); 3Institute of Enzymology, HUN-REN Research Centre for Natural Sciences, 1117 Budapest, Hungary; spisak.sandor@ttk.hu

**Keywords:** *mcr-1 Escherichia coli*, colistin resistance, waterfowl, plasmid genome, core genome MLST, multiresistance, whole-genome sequencing

## Abstract

Plasmids carrying high-risk resistance mechanisms in pathogenic *E. coli* have gained particular attention in veterinary medicine, especially since the discovery of the colistin resistance gene, *mcr-1*. Here, we provide the first evidence of its emergence and describe the complete *mcr-1* plasmid sequence of a multi-resistant avian pathogenic *E. coli* (APEC) strain from waterfowl in Hungary. Whole-genome sequencing analysis and core-genome MLST were performed to characterize the genome structure of the *mcr-1* plasmid and to reveal the phylogenetic relation between the Hungarian duck strain Ec45-2020 and the internationally circulating *mcr-1*-positive *E. coli* strains from poultry and humans. Results showed that plasmid pEc45-2020-33kb displayed a high level of genome identity with *mcr-1* plasmids of IncX4 type widespread among human, animal and food reservoirs of enteric bacteria of public health. The *mcr-1*-positive *E. coli* strain Ec45-2020 belongs to the ST162 genotype, considered as one of the globally disseminated zoonotic genotypes of MDR *E. coli*. In accordance with international findings, our results underline the importance of continuous surveillance of enteric bacteria with high-risk antimicrobial resistance genotypes, including neglected animals, such as waterfowls, as possible reservoirs for the colistin resistance gene *mcr-1*.

## 1. Introduction

*Escherichia coli* is a ubiquitous bacterium commonly found in the gastrointestinal tract of warm-blooded animals and a key bacteria of their normal intestinal microbiota. However, pathogenic strains of *E. coli* can infect animals and humans, causing various intestinal or extraintestinal disorders depending on their virulence factors [1]. In the poultry industry, Avian Pathogenic *E. coli* (APEC) is a concern, accounting for serious economic losses through elevated mortality rates across different age cohorts of birds and the resulting carcass condemnations at slaughterhouses [2]. Recent findings indicate a remarkable genetic overlap between APEC and human Extraintestinal Pathogenic *E. coli* (ExPEC), which confers APEC the ability of zoonotic transmission through the food chain [3,4]. APEC strains, like other pathogenic *E. coli*, are originated and selected from commensal *E. coli* strains by harboring a wide range of virulence genes to increase their pathogenic potential [5,6,7].

Moreover, *E. coli* strains frequently carry antimicrobial resistance genes (ARGs) encoding for multiresistance (MDR), hampering the success of infection treatment. Furthermore, such strains of poultry can facilitate the spread of antibiotic resistance to humans, especially through the consumption of poultry meat [6]. Antimicrobial resistance (AMR) plasmids and horizontal gene transfers are key drivers of the spread of AMR, while the clonal spread of ARG is of secondary importance. AMR/MDR plasmids have been gaining increased attention in veterinary medicine, especially since the discovery of the plasmid-encoded high-risk gene *mcr-1*, which encodes for colistin resistance [8]. This first detection of *mcr* genes changed the attitude toward colistin which had been thought safe in therapies because of the absence of evidence for ARG shifting. Colistin is one of the last resort antimicrobial agents in human medicine with quick, bactericidal effects and is still used widely for the therapy of selected enteric diseases in farm animals [9]. This practice was based on the chromosomal-encoded *mcr* gene, which is considered to pose minimal public health risks due to its slow transmission rates. However, the discovery of *mcr* genes requires re-evaluation, as several *mcr* alleles (*mcr-1* to *mcr-10*) have been identified in various bacterial species and multiple host sources [10].

Traditional serotyping and Multi-Locus Sequence Typing (MLST) are considered to be the gold standards for epidemiological investigation and comparative genotyping of *E. coli* strains. However, they provided limited resolution and depth. The advent of Next-Generation Sequencing (NGS) has revolutionized genotyping and enabled high-throughput genome comparison. Core genome Multi-Locus Sequence Typing (cgMLST) enables high-resolution genotyping based on polymorphism of 1500 to 4000 instead of the seven housekeeping genes of the MLST scheme [11]. Therefore, cgMLST has become a widely accepted technique for deep-genotyping of several enteric bacteria, including *E. coli*.

So far, the emergence of colistin-resistant *E. coli* strains carrying the *mcr-1* gene has been reported in Hungary in healthcare settings [12], but it has not yet been detected in animals or animal-related products. To our knowledge, this is the first report of the emergence and whole-genome sequence-based characterization of a *mcr-1*-positive *E. coli* originating from waterfowl in Hungary.

## 2. Results

### 2.1. Isolation and Identification of the Colistin Resistant E. coli Strain Ec45-2020

Overall, 479 *E. coli* strains were isolated from 483 caecal/cloacal samples representing five species of poultry from 34 farms and four slaughterhouses (Table 1) [13,14]. Remarkably, the *mcr-1* gene was only detected in one *E. coli* strain by PCR, and its presence was confirmed using Sanger sequencing. Designated as Ec45-2020, this *mcr-1*-positive APEC strain was isolated from a six-day-old duckling that died from a possible systemic infection on the farm and was transported to the National Diagnostic Directorate (Nébih-ÁDI), Budapest, for diagnostic investigations. Strain Ec45-2020 displayed the Amp-Chl-Cip-Col-Sul-Tet-Tmp resistance phenotype, and its MIC for colistin was 8 µg/mL. The serotype of the strain was identified as H10:O55.

### 2.2. Phylogenetic Relation and Genomic Diversity of mcr-1-Positive E. coli Strains from Poultry and Humans

To reveal the phylogenetic relation and genomic diversity among *mcr-1*-positive *E. coli* strains, a selected online collection of strains was assembled based on a targeted search within the Bioinformatics Virtual Bioresource Center (BV-BRC) database. The sequence type (ST) of the strains was identified using MLST, based on the available whole-genome sequences. The BV-BRC database search yielded 114 *mcr-1*-positive *E. coli* strains from poultry and 390 from humans, isolated between 2016 and 2020 (Appendix A). The collection represented a global distribution with strains originating from 17 countries, where 81.1% of human and 58% of poultry strains have been isolated in China. Among the 114 poultry strains, 2 were specifically isolated from ducks: 1 from China and the Hungarian strain Ec45-2020 under study.

The tested *mcr-1*-positive *E. coli* strains were allocated into 192 STs, the majority of them represented individual genetic lineages with one isolate each (Appendix A). In total, 28 STs were regarded as overlapping between the poultry and human strains of *mcr-1*-positive *E. coli*. Of these, ST10, ST48, ST101 and ST206 were most frequently (7.1–23.9%) identified, but predominantly characterizing the human strains (Table 2).

In addition to the commonly detected STs, only a limited number of poultry-specific STs have been identified. These included ST351, ST3941 and ST6751, which were identified solely in chickens and exhibited international distribution (Figure 1). The Hungarian strain Ec45-2020 was identified as ST162, a sequence type that was also identified in one chicken and two human strains originating from China (Figure 1 and Appendix A).

To provide a deeper understanding of genomic diversity, cgMLST analysis of 238 *mcr-1*-positive *E. coli* strains, selected to represent ST lines common between poultry and humans (Table 2), was performed based on the polymorphism of 2138 genes of the core genome by using the strain *E. coli* K-12 MG1655 as a reference.

The core genome phylogenetic tree displayed substantial diversity among the poultry and human genomes of the *mcr-1*-positive *E. coli* (Figure 2). Overall, three major clusters and numerous subgroups could be distinguished. Strains of ST48 and ST206 clustered together with ST10, exhibiting the most significant genomic diversity. The Hungarian duck ST162 strain Ec45-2020 shared identical cgMLST genotypes with Chinese human strains, but differed significantly from the Chinese *mcr-1*-positive *E. coli* strain of duck origin.

### 2.3. Genomic Determinants of Antimicrobial Resistance and Virulence

A whole-genome sequence-based in silico prediction of acquired resistance and virulence genes was performed to reveal the genomic environment of the colistin resistance gene *mcr-1*. As a result of the whole-genome analysis and contig assembly, a chromosomal contig of 4,966,963 bp (GenBank Acc. no. CP134085) and the coexistence of five circular plasmids were identified in the duck *E. coli* strain Ec45-2020. The chromosome displayed typical features of the APEC genotype, including virulence genes (VGs) such as *astA* (heat-stable enterotoxin), *fyuA* (siderophore receptor), *hlyE* (hemolysin) and *lpfA* (long polar fimbriae). AMR phenotyping revealed a high level of multi-resistance in the strain, including resistance to ampicillin, chloramphenicol, ciprofloxacin, colistin, sulfonamides, tetracycline and trimethoprim, as conferred by a large set of antimicrobial resistance genes (ARGs). These ARGs were located on three different plasmids ranging in size from approximately 33 to 255 kb.

The *mcr-1* gene was located on a 33,541 bp plasmid designated pEc45-2020-33kb (CP134089). In silico prediction of ARGs, VGs and plasmid replicon types revealed that this plasmid is of IncX4 type and exclusively harbors the *mcr-1* gene (Figure 3). A larger MDR plasmid, pEc45-2020-254kb (CP134088), was characterized as an IncH replicon type, conferring resistance to trimethoprim (*dfrA12*), aminoglycosides (*aadA1,2*), sulfonamides (*sul3*), fluoroquinolones (*qnrS1*) and phenicols (*cmlA1*/*floR*) (Appendix A).

The virulence genotype of the strain Ec45-2020 partially relies on a high number of VGs, carried by the plasmid pEc45-2020-190kb (CP134087). Of these, genes *hly*, *tra*, *iut* and *iuc* are underlined as they are involved in hemolysis, conjugative plasmid transfer and iron uptake. This plasmid also carried clinically important ARGs, thereby functioning as a hybrid plasmid. Remarkably, the gene cluster *bla*_TEM-135_-*sul*2-*tet*(A),(M), was identified in duplicate within the Ec45-2020 genome, contributing to resistance against ampicillin, sulfonamides and tetracyclines (Figure 3).

In contrast to the aforementioned resistance and virulence plasmids, the two remaining plasmids of the strain, pEc45-2020-5kb (CP134090) and pEc45-2020-101kb (CP134086), did not carry any determinants encoding for resistance and/or virulence (data not shown).

### 2.4. Genome Architecture of the mcr-1 Plasmid pEc45-2020-33kb Identified in Hungary

To understand the host and source diversity associated with plasmids carrying the *mcr-1* gene, a BLASTn analysis was conducted. This analysis revealed 26 plasmid sequences with a 100% query coverage and a pairwise identity ranging between 93% and 98% when mapped to pEc45-2020-33kb. For comparative genomics, nine plasmids were selected, each representing different bacterial species and plasmid genome structures (Figure 4). The analysis indicated that the backbone of the *mcr-1*-bearing IncX4 plasmids, similar in size to pEc45-2020-33kb, was highly conserved across different enterobacteria species, including *E. coli*, *K. pneumoniae* and *E. fergusonii*.

The sequence comparison showed that the backbone of *mcr-1* IncX4 plasmids, the size of pEc45-2020-33kb, is highly conserved in different enterobacteria. Notably, these plasmids showed a high level of similarity in the region encoding the Type IV Secretion System (T4SS) components, specifically between the *virD4* and *taxC* relaxase genes, as well as in the region harboring the *mcr-1* gene, located between the replication origin and the mobile element IS6. In addition to IS6, other IS elements such as IS66 and IS3 were also found in the genomes of some *E. coli* plasmids. Similarly, the beta-lactam resistance gene *bla*_TEM-135_ was also identified as a second resistance gene in the plasmid pEC200574.

Although putative conjugative elements of the T4SS and the relaxase gene were also identified by the *oriT*finder [15], the *oriT* region itself remained undetectable. Moreover, the plasmid pEc45-2020-33kb was found to be non-transferable under the experimental conditions we used for conjugative transfer analysis.

## 3. Discussion

Resistance to polymyxins, including colistin in Gram-negative enteric bacteria has been very uncommon, and the mechanism was mainly conferred by mutations in the chromosomal genes *mgrB*, *phoP*/*phoQ*, and *pmrA*/*pmrB*, inducing modification of bacterial LPS and protection against these cationic peptides [16]. Thus, for several decades, colistin and polymyxins have been and still are widely and successfully used in veterinary medicine against these Gram-negative infections.

This favorable situation has changed since the discovery and spread of the plasmid-mediated colistin resistance gene *mcr-1* [8], resulting in increasingly occurring resistance to these cationic peptides worldwide. Therefore, in most EU countries, including Hungary, only the restricted clinical use of colistin is permitted for veterinary practice and it has been used at an essentially decreasing rate during the last few years [17,18]. However, the strict regulation on antibiotic (especially on colistin) treatment in farm animals is far from being global and several antibiotics, including colistin, are used commonly and maybe preventive (prophylaxis or meta-phylaxis) outside the EU, and the ARGs, especially *mcr-1* genes, are getting widely distributed between animals and the environment [19,20,21,22,23,24].

Our findings regarding the very low frequency of *mcr-1*-positive *E. coli* (1 out of 479) are in accordance with the aforementioned EU data on the decreasing trend of colistin resistance. Furthermore, a recent independent study is in line with our data, indicating a lack of *mcr-1*-positive *E. coli* strains related to Hungarian turkey imports [22]. The occurrence of transferable colistin resistance was reported earlier only in human patients of a tertiary care center in Hungary [12]. The results of a systematic screening and detailed analysis of the resulting *mcr-1*-positive *E. coli* of animal (poultry) origin are reported for the first time in Hungary and form the basis of this study.

Chicken and waterfowl products are considered important meat sources in several countries, which urges for the intensive and regular surveillance of antimicrobial resistance in broilers and waterfowls, which has not yet been included in the compulsory EU surveillance systems [21]. In comparison to chicken, however, limited studies address the molecular epidemiology of *mcr-1*-positive *E. coli* from ducks. Therefore, we believe our study will add a valuable contribution to the field. Here, we provide the first evidence of its emergence and describe the complete *mcr-1* plasmid sequence of a multi-resistant *E. coli* strain of poultry origin in Hungary. Whole-genome-based epidemiological relations between the Hungarian duck strain Ec45-2020 and internationally circulating chicken and human *mcr-1*-positive *E. coli* strains isolated between 2016 and 2020 were also revealed.

Since its first emergence [8], an increasing number of studies are reporting the alarming prevalence of *mcr-1*-positive *E. coli,* not only alone but also co-harboring CTX-M type ESBL- and NDM carbapenemase genes from multiple animal sources, including ducks [25,26], but also from the natural environment, posing a serious concern from the One Health perspective [23,27]. The diversity is similarly high considering the type and genome structure of the corresponding *mcr-1* plasmids. Of them, the IncX4 and IncI2 plasmids are the key drivers of the worldwide spread of the *mcr-1* gene in several *Enterobacterales* species of top public relevance, including ESKAPE pathogens and high-risk MDR clones of *E. coli* and monophasic *S*. Typhimurium [20,23,28].

The IncX4 plasmid pEc45-2020-33kb identified here in a *mcr-1*-positive *E. coli* strain of duck origin showed a high level of genome identity with other IncX4 plasmids widespread among human, animal and food reservoirs of enteric bacteria of public health relevance. IncX4 plasmids carrying the *mcr-1* gene were usually reported as transferable under laboratory conditions [29,30]. Contrary to our expectations, we found that the plasmid pEc45-2020-33kb was not transferable under the applied experimental conditions. Self-transmissible or conjugative plasmids require the presence of a four-component conjugative apparatus: a transfer origin *oriT*, a relaxase, a type IV coupling protein, and a T4SS [31]. However, the *oriT* was not detectable in the sequence of plasmid pEc45-2020-33kb, which could be a possible reason why this plasmid was not conjugable. Considering that the *mcr-1* plasmid coexists with two other MDR plasmids in the strain Ec45-2020, we speculate a potential conjugation competition among these resistance plasmids, which may impede the transmission of the pEc45-2020-33kb plasmid. This interesting aspect of plasmid interaction warrants further investigation.

The *mcr-1*-positive *E. coli* strain Ec45-2020 belongs to the ST162 genotype, considered as one of the globally disseminated genetic lineage of MDR *E. coli*, also comprising pathogenic (e.g., APEC) and commensal strains from multiple sources in poultry and humans [32,33,34,35]. Recently, ST162 has become an *mcr-1*-bearing *E. coli* genotype in poultry and wild avian species, posing a risk to food safety [19,36]. Moreover, recent findings indicate, that ST162 avian *mcr-1*-positive *E. coli* isolates exhibit high pathogenicity, causing bloodstream infections in animal models [37]. Although, apart from the *mcr-1* gene, the pEc45-2020-33kb plasmid does not carry other resistance and virulence genes, the ST162 genotype and the co-existence of MDR/virulence plasmids potentiate the zoonotic and public health importance of the host *E. coli* strain Ec45-2020.

A comprehensive molecular epidemiological study on the population structure of avian *mcr-1*-positive *E. coli* underlined the relevance of ST10, ST101 and ST162 as popular genotypes associated with the *mcr-1* gene in chickens and ducks in China [38]. In addition, we found that ST206 is also important in this context, being detected at the same prevalence in poultry and humans. Similarly, according to a recent study by Wang et al. (2021) [26], ST156 and ST354 could also be considered prevalent *mcr-1*-positive *E. coli* genotypes in duck populations.

The phylogenomic comparison revealed that the genotypes of 238 strains of *mcr-1*-positive *E. coli* STs identified as common between poultry and humans were highly heterogeneous and independent of host origin. This suggests that an extreme diversity of *E. coli* genotypes with zoonotic potential could potentially be involved in the acquisition and dissemination of resistance to the last-resort antibiotic colistin.

## 4. Materials and Methods

### 4.1. Collection of E. coli Strains Subjected to mcr-1 Screening

Tested *E. coli* strains have been isolated from previously reported samples (n = 483), collected from slaughterhouses (chicken, turkey, duck and goose) and small-scale farms (chicken and pigeon), but samples provided by the National Diagnostic Directorate (Nébih-ÁDI, Budapest, Hungary) between 2016 and 2020 were also included [13,14]. Caecum and culled animal carcasses in slaughterhouses and caecum in Nébih-ÁDI or cloacal swab tampons in small-scale farms (from live birds) were used for the isolation of *E. coli* strains.

The same isolation method was applied to all samples. Firstly, the collection swabs were immersed in LB (Luria-Bertani) broth to overnight culture at 37 °C Then, they were inoculated on MacConkey selective media and were incubated for 24 h at 37 °C. Lactose-positive colonies were purified with serial reinoculation on MacConkey agar. Lactose-negative colonies were used for further identification processes when only lactose-negative colonies were visible after the first smearing. All isolated and purified bacteria were confirmed firstly with primary and secondary biochemical tests (catalase, oxidase and indol, urease test) as *E. coli*. They were verified later with PCR tests as *E. coli* with primers (EC_F: CCAGGCAAAGAGTTTATGTTGA EC_R: GCTATTTCCTGCCGATAAGAGA; product 212 bp) [39]. Pure cultures of bacteria were frozen after their isolation at −80 °C with 15% glycerin.

### 4.2. Identification of the mcr-1 Gene and Antibiotic Susceptibility Testing

For the detection of the *mcr-1* gene, DNA was extracted from an overnight culture of the aforementioned *E. coli* strains by boiling [14]. Then, the templates were used for PCR assay which was prepared from DreamTaq polymerase (1.25 U; Thermo-Scientific, Waltham, MA, USA), DreamTaq green buffer (5 µL; Thermo-Scientific), dNTP (0.2 mM each; Thermo-Scientific), *mcr-1* primer pair (CLR5F: 5′-CGGTCAGTCCGTTTGTTC and CLR5R: 5′-CTTGGTCGGTCTGTAGGG; 0.5 µM each; Merck Life Science, Burlington, MA, USA) and bidistilled water in 25 µL final volume [8]. PCR reaction conditions were: 1 min at 94 °C, 1 min at 94 °C, 30 s at 51 °C, 30 s at 72 °C and final extension for 5 min at 72 °C; the 2nd–4th steps were repeated in 25 cycles [8]. Amplicons were separated in 1.5% agarose gel (110 V, 45 min) and Sanger sequenced by BaseClear B.V. (Leiden, The Netherlands). All Sanger sequences were verified by BLAST.

The resistance phenotype was determined using disc diffusion against the following antibiotic compounds: ampicillin (AMP10), cefotaxime (CTX5), chloramphenicol (CHL30), ciprofloxacin (CIP5), gentamicin (GEN10), meropenem (MEM10) nalidixic acid (NAL30), sulphonamide compounds (SUL300), tetracycline (TET30) and trimethoprim (TMP5). The colistin-resistant phenotype was confirmed by determining the minimum inhibitory concentration (MIC). Antibiotic susceptibility testing was performed according to the guidelines and interpretation criteria of the European Committee on Antimicrobial Susceptibility Testing [40]. The intermediate category was interpreted as susceptible. *E. coli* strain ATCC 25922 and was used as a reference.

### 4.3. Whole-Genome Sequencing and Analysis

Whole-genome sequencing of the *E. coli* strain Ec45-2020 was performed by Eurofins Biomi Ltd. (Gödöllő, Hungary). Short reads of 2 × 250 bp were generated on the Illumina MiSeq sequencing platform using v2 sequencing chemistry. To enable the reconstruction of plasmid sequences, the strain was sequenced in parallel on the MinION platform using an ONT MinION Mk1C device.

Quality analysis of the Illumina raw reads was performed using the built-in read-processing pipeline of Geneious Prime^®^ v2023.1.2 (Biomatters Ltd., Boston, MA, USA). Accordingly, BBDuk was used for trimming and BBNorm for error correction and normalization of the read sequences.

Nanopore sequencing data underwent highly accurate basecalling using Guppy_basecaller v6.3.8+d9e0f64, followed by merging of the successfully processed reads. Subsequently, the Porechop tool v0.2.4 and Nanofilt v1.16 were used to eliminate sequencing adapters and read trimming to enhance read quality. The resultant cleaned raw read FASTQ file was then put into the Unicycler program v0.5.0 to facilitate the identification and assembly of both circular and linear contigs representing the chromosomes and plasmids, respectively. Fine-tuning of assembled genome sequences was performed using Nanopolish v0.14.0, enabling the generation of a polished genome with high-quality variant calls. Polished genome sequences were then used as a reference for Illumina read mapping, and the resulting consensus sequences were manually cured and finalized to produce the complete genome of strain Ec45-2020 suitable for in silico WGS analysis. Genome annotation was primarily performed with the RAST (Rapid Annotations using Subsystems Technology) server v2.0 [41]. However, complete *mcr*-plasmid genome sequences were also considered for refinement. Chromosomal and plasmid genome sequences of strain Ec45-2020 were deposited in the BioProject PRJNA1012593.

### 4.4. Core Genome-Based Multi-Locus Sequence Typing (cgMLST)

To reveal the genomic diversity and phylogenetic relation among *E. coli* strains carrying the *mcr-1* gene, cluster analysis of a large set of *E. coli* genomes was performed on the basis of the core genome genes. The *mcr-1*-positive *E. coli* strains of poultry and human origin, isolated between 2016 and 2019, were selected from the database of the Bacterial and Viral Bioinformatics Resource Center (BV-BRC) (https://www.bv-brc.org/; accessed on 12 June 2023). Only E. coli strains with high-quality whole-genome sequencing data and with known sequence types (STs) were subjected to cgMLST analysis using the Ridom SeqSphere+ software v9.08 [42].

First, sequence types (STs) were determined by MLST based on the polymorphism of the seven housekeeping genes according to the Warwick MLST scheme for *E. coli* [43]. Then, the core genes of *E. coli* were analyzed by blasting all genome sequences against the E. coli reference strain K-12 MG1655 (GenBank accession no. NC_000913). The serotype of the strain Ec45-2020 was identified using SerotypeFinder 2.0 [44].

### 4.5. In Silico Analysis of the Antibiotic Resistance and Virulence Genotypes and Testing the Transferability of the mcr-1 Plasmid

Antimicrobial resistance and virulence genes (ARGs and VGs) were identified using appropriate web-based typing tools of the Center for Genomic Epidemiology (CGE) platform (https://www.genomicepidemiology.org/services/; accessed on 10 July 2023). According to this, ResFinder 4.1 [45], VirulenceFinder2.0 [46] and PlasmidFinder 2.1 [47] were used for the in silico detection of plasmid-borne antibiotic resistance and of virulence genes and for the typing of plasmids based on the replicon type. The assembled plasmid and chromosomal contigs were used as inputs, while the minimum thresholds for sequence identity and length coverage were set to 90% and 80%, respectively.

To study the transferability of the *mcr-1* plasmid, conjugation experiments were performed using the rifampicin-resistant *E. coli* K-12 C600 as the recipient strain. For this, overnight LB broth cultures of Ec45-2020 donor (D) and recipient (R) strains were 50× diluted and incubated for 2 h (37 °C, 130 rpm) to obtain fresh LB cultures in the log growth phase. Next, the OD600 of both cultures was adjusted to 0.3, and individual cultures were mixed at a D:R ratio of 1:1.

The conjugation potential was tested at 1, 2 and 4 h post co-incubation on both the liquid and solid media. The incubation temperature was set to 37 °C (adapted to the plasmid incompatibility group), following the methodology of Rodriguez-Grande and Fernandez-Lopez (2019) [48]. For enumeration of donor, recipient and transconjugant colonies, 10× serial dilutions (100-10-6) of the conjugation mixtures were prepared and droplet-plated onto LB plates supplemented with the appropriate antibiotics; colistin (5 µg/mL) for D, rifampicin (200 µg/mL) for R, and a combination of colistin/rifampicin for the selection and enumeration of potential transconjugants. Conjugation experiments were performed twice, with 6 replicates each.

## 5. Conclusions

This study provides the first evidence of its emergence and describes the complete *mcr-1* plasmid sequence of a multi-resistant ST162 APEC strain from a duck in Hungary. The comprehensive phylogenomic analysis highlighted the extremely high genotype diversity, but the overlapping clones of colistin-resistant *E. coli* between poultry and humans. Plasmid pEc45-2020-33kb displayed a high level of genome identity with *mcr-1* plasmids of IncX4 type of global distribution in enteric bacteria from humans and multiple animal sources. In accordance with international findings, our results underline the importance of continuous surveillance of enteric bacteria with high-risk antimicrobial resistance genotypes, including neglected animals such as waterfowls as a possible reservoir for the colistin resistance gene *mcr-1*.

## Figures and Tables

**Figure 1 antibiotics-12-01519-f001:**
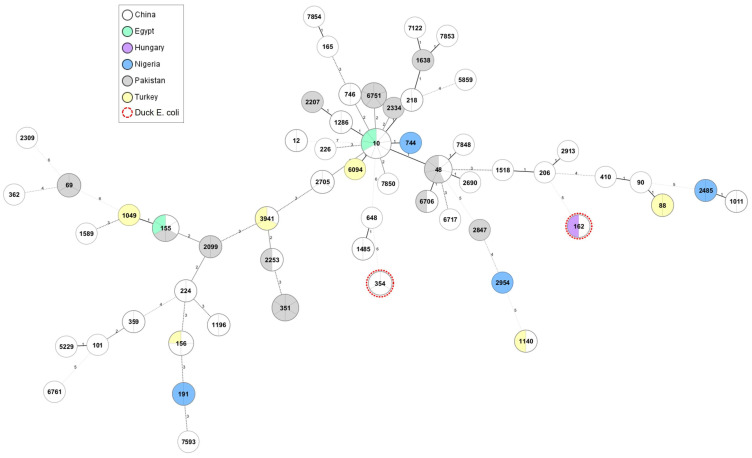
Phylogenetic diversity and geographical distribution of *mcr-1*-positive *E. coli* strains with origin from poultry (2016 and 2020). The Minimum Spanning Tree was generated to represent the phylogenetic relation among 114 *E. coli* strains from poultry based on the ST type, namely on the polymorphism of seven housekeeping genes. Nodes containing multiple strains are demarcated by grey lines. Red dotted circles indicate the STs corresponding to the duck isolates of *mcr-1*-positive *E. coli*. Distance lines change from black to dotted grey as the phylogenetic distance between the strains increases. The thickness of the distance line is inversely proportional to the distance value.

**Figure 2 antibiotics-12-01519-f002:**
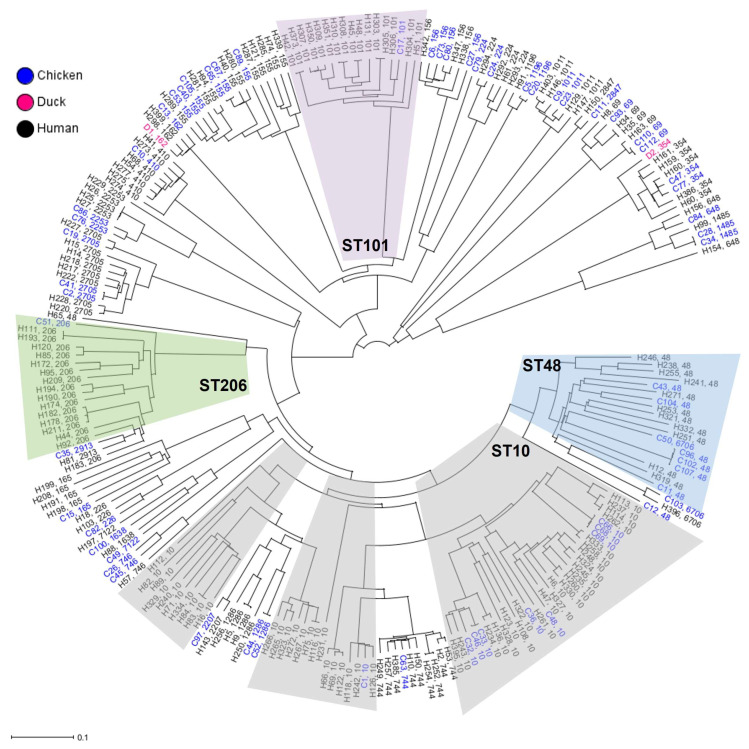
Core genome diversity of common ST lines of *mcr-1*-positive *E. coli* from poultry and humans. The Neighbor Joining Tree showing the genomic diversity of 238 *E. coli* strains was calculated based on the polymorphism of 2138 target genes of the core genome. Core genes were identified by blasting all genome sequences against the reference strain *E. coli* K-12 MG1655 (GenBank accession no. NC_000913).

**Figure 3 antibiotics-12-01519-f003:**
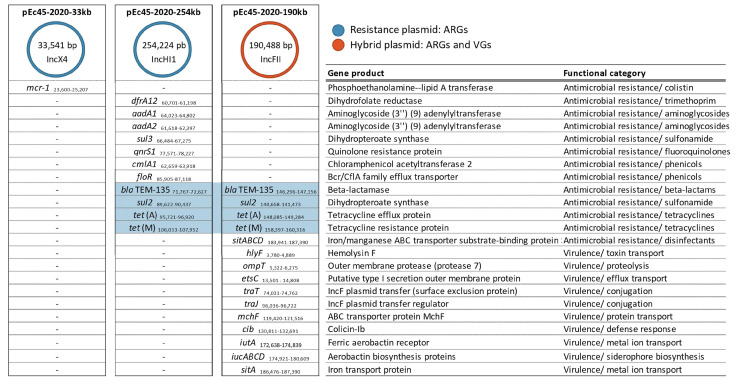
Genomic features of acquired resistance and virulence and their distribution among the three resistance and hybrid plasmids. The sequence positions of the listed genes are indicated by subscripts.

**Figure 4 antibiotics-12-01519-f004:**
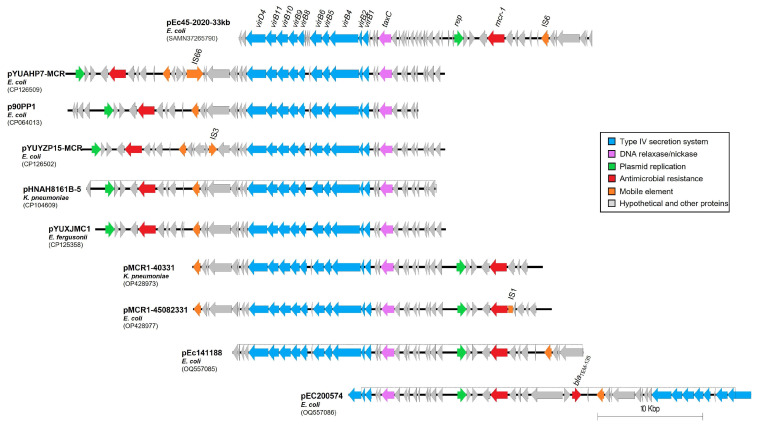
Comparison of complete linear plasmid sequences to pEc45-2020-33kb. The plasmid set was selected from the top 100 BLAST hits which showed 100% query coverage and >95% pairwise identity. For each strain, GenBank accession numbers are indicated in parentheses.

**Table 1 antibiotics-12-01519-t001:** Distribution of *E. coli* strains (n = 479) subjected to *mcr-1* screening according to sampling environment and host species.

	Pigeon	Chicken	Turkey	Goose	Duck
Number of sampled farms (dead animals)	5 (n = 38)	8 (n = 27)	1 (n = 4)	9 (n = 53)	8 (n = 36) *
Number of sampled slaughterhouses (culled animals)	-	1 (n = 132)	1 (n = 51)	1 (n = 48)	1 (n = 51)
Number of sampled outdoor keeping farms (live animals)	-	1 (n = 18)	1 (n = 6)	-	-
Number of sampled backyard farms (live animals)	-	1 (n = 15)	-	-	-

* Origin of the *mcr-1*-positive sample.

**Table 2 antibiotics-12-01519-t002:** Sequence types of *mcr-1*-positive *E. coli* strains showing overlap between poultry and humans.

Genotype	Poultry (%)	Human (%)	Total % (no.)
ST10	13.85	73.85	23.95 (57)
ST48	10.77	18.46	7.98 (19)
ST101	1.54	24.62	7.14 (17)
ST206	1.54	24.62	7.14 (17)
ST155	9.23	15.38	6.72 (16)
ST2705	4.62	12.31	4.62 (11)
ST744	1.54	13.85	4.20 (10)
ST354 *	4.62	7.69	3.36 (8)
ST410	1.54	10.77	3.36 (8)
ST69	4.62	6.15	2.94 (7)
ST156	6.15	4.62	2.94 (7)
ST224	3.08	6.15	2.52 (6)
ST1286	3.08	6.15	2.52 (6)
ST1011	3.08	6.15	2.52 (6)
ST2253	3.08	6.15	2.52 (6)
ST165	1.54	6.15	2.10 (5)
ST162 *	3.08	3.08	1.68 (4)
ST226	1.54	3.08	1.26 (3)
ST648	1.54	3.08	1.26 (3)
ST746	3.08	1.54	1.26 (3)
ST1196	3.08	1.54	1.26 (3)
ST1485	3.08	1.54	1.26 (3)
ST6706	3.08	1.54	1.26 (3)
ST1638	1.54	1.54	0.84 (2)
ST2207	1.54	1.54	0.84 (2)
ST2847	1.54	1.54	0.84 (2)
ST2913	1.54	1.54	0.84 (2)
ST7122	1.54	1.54	0.84 (2)

* Sequence type of the duck *E. coli* isolates.

## Data Availability

The datasets generated for this study are available in BioProject PRJNA1012593.

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
