# Peer review of "Emergence and Genomic Features of a mcr-1 Escherichia coli from Duck in Hungary"

_antibiotics, 2023, doi:10.3390/antibiotics12101519_

Round 1

Reviewer 1 Report

This manuscript describes a study of E. coli isolated from birds and focuses on a single isolate from a duck that has an mcr-1 gene and an APEC plasmid. The authors did WGS analysis and assembled a closed genome and plasmids by combining Illumina and Nanopore sequence data. To determine the context of this single isolate found in a Hungarian duck, the selected WGS data from multiple isolates and species of bacteria carrying these genes. This comparison was fairly detailed, and the results were well presented in the text and the figures. Overall, this is an interesting study and is well written except for the discussion section. The discussion section seems to have the concentration of English usage errors and many confusing sentences and statements. In some cases, the reader cannot determine what the authors were trying to communicate. A careful proofreading and correction of the English usage and typos would greatly improve this section of the paper. Specific comments the authors could address follow:

1. 72-75, awkward sentence please rewrite

2. 155, the ARGs are not randomly distributed, they are just not organized into a single cassette on a single plasmid, please reword

3. 157-158, delete "of in size"

4. 175, the plasmid is not empty, it just lacks ARG and VG

5. 202-205, this statement is very confusing. Do you mean that intrinsic resistance kept bacteria from acquiring mcr-1, or do you mean it wasn't detected because researchers thought the resistance was intrinsic?

6. 208, change have to has and add in before the an.

7. 215, by preventative, do you mean use prophylactically or as meta-phylaxis?

8. 215, "are getting were" do you mean worse?

9. 229, add , after Therefore

10. 255, replace with address, the way it is written it sounds like you're planning to send a letter :-)

11. 264, these plasmids are not naked, they just lack ARG and VG

12. You could consider adding a conclusion section to summarize your findings

The discussion section requires attention to the quality of English usage.

Reviewer 2 Report

In this study, the authors report for the first time the carriage of a non-transferable plasmid harboring the mcr-1 gene carried by a multiresistant avian pathogenic E. coli (APEC) strain from waterfowl in Hungary. They used a whole genome sequencing analysis and core-genome MLST to characterize the genome structure of the mcr-1 plasmid and to reveal the phylogenetic relation between the Hungarian strain and the internationally circulating mcr-1-positive E. coli strains from poultry and humans.

This is a well-written study that provides very relevant information on the emergence of the mcr-1 gene

However, the entire description of the number of strains not carrying this gene appears completely unnecessary for this article, and I suggest that it be removed from the manuscript to focus on the strain carrying the mcr-1 gene. Thus, I recommend removing Table 1 and changing the section 4.1 of the Materials and Methods and changing paragraph between lines 71-86 to maintain only information of E. coli Ec45-2020. In addition, Supplementary file should be removed.

Another aspect to be corrected corresponds to the determination of the MIC values ​​of the strains used in the conjugation assays, as described in lines 303-306: “In addition, the minimum inhibitory concentrations (MICs) of the donor, recipient, and transconjugant strains were determined for ampicillin, chloramphenicol, ciprofloxacin”. These results are not included in the manuscript, but they are not important considering that plasmid pEc45-2020 that harbors the mcr-1 gene was not able to be transferred, thus I suggest removing this paragraph.

Furthermore, it is highly suggested to include a supplementary file showing the identity percentages of the ARGs reported, because many times ARGs are wrongly reported despite the genes have a very low similarity to the ARG.

Minor remarks

Line 294: Please change “Escherichia coli” to read “Escherichia coli

Reviewer 3 Report

The data included in this paper are interesting for the area of colistin resistance. Authors present a complete analysis of the mrc-1 plasmid which allowed them to compare this strain to others isolated around the world and to reported the non-transferable nature of the plasmid. I have only to minor comments:

in line 127 please include the criteria used to select the 238 mrc-1 positive strains

lines 191-193: these elements were not in the duck mrc-1 plasmid, is there any reason to highlight their presence in the other plasmids?

In the title: i would stress that the study refers to one specific E. coli strain, so it would be:  Emergence and genomic features of a mcr-1 Escherichia coli strain  from duck in Hungary

For me the language English quality is OK
